# Influence of Satellite Motion Control System Parameters on Performance of Space Debris Capturing

**Mahdi Akhloumadi [1] and Danil Ivanov [2],*** 

1    Moscow Institute of Physics and Technology, State University, Institutsky Lane 9, Dolgoprudny, 141701 Moscow Region, Russia; akhloumadi@phystech.edu
2    Keldysh Institute of Applied Mathematics RAS, Miusskaya sq. 4, 125047 Moscow, Russia
*    Correspondence: danilivanovs@gmail.com

**Abstract:** Relative motion control problem for capturing the tumbling space debris object is considered. Onboard thrusters and reaction wheels are used as actuators. The nonlinear coupled relative translational and rotational equations of motion are derived. The SDRE-based control algorithm is applied to the problem. It is taken into account that the thrust vector has misalignment with satellite center of mass, and reaction wheels saturation affects the ability of the satellite to perform the docking maneuver to space debris. The acceptable range of a set of control system parameters for successful rendezvous and docking is studied using numerical simulations taking into account thruster discreteness, actuators constrains, and attitude motion of the tumbling space debris.

**Keywords:** formation flying; relative motion control; space debris capture; SDRE-based control; reaction wheels saturation; thruster misalignment

## 1. Introduction

Further exploration of near-Earth space in the short term will become impossible without solving the problem of space debris removal. A number of international projects are aimed at its solution [1,2]. Space debris removal approaches can be divided into two classes: passive and active [3–5]. In the case of passive approach the space debris are removed with the help of external natural forces, for example, aerodynamic drag in low Earth orbits [6,7], solar pressure [8], ionospheric drag [9] etc. Active removal of inactive spacecrafts and rocket stages often involves the use of special small satellites that can attach themselves on the space debris object or capture it with a manipulator [10], net [11,12] or harpoon with tether [13,14], and change its orbit using the on-board motion control system. Active debris removal implies autonomous relative motion control in order to achieve relative state vector required for capturing. An onboard propulsion is often considered for the translational motion control and reaction wheels are for the attitude maneuvers. However, small satellites have restrictions on mass, size, energy, on-board computing power, and the composition of the control system equipment, which complicates the on-board control algorithms for the main modes of motion at the mission design stage, taking into account the limited capabilities of the satellites. The reaction wheels may experience saturation which must be avoided, the saturation may be caused by thruster misalignment or by high angular velocity of the tumbling debris. These situations may lead to the failure of the object capturing. That is why it is required to study the influence of the control system parameters on rendezvous and capturing maneuver capabilities.

The problem of relative translational and attitude motion control is well studied and a big variety of control approaches is developed. For example, sliding control-based algorithms [15,16] are

developed for the relative orbit-attitude tracking problem, for the rendezvous problem the swarm particle optimization algorithm is applied for the required trajectories generation [17], for the docking stage with non-cooperative object the majority of the proposed control algorithms are fuel-optimal or time-optimal [18–22]. The fuel-optimal trajectory generation algorithms are computationally intensive, its implementation in a real-time system is very challenging. Therefore, the optimal algorithms for calculating trajectories are often replaced by computationally simpler and faster non-optimal ones [23,24], however their performance is strongly dependent on initial conditions at the docking stage. As a compromise between two approaches a feedback control law developed for minimization of some defined cost-function can be applied to the problem. A linear quadratic regulator (LQR) is a well-known example of such an algorithm, though the relative motion equations are highly non-linear. To overcome this inconsistency the motion equations are linearized in the vicinity of the current state vector and LQR-like State-Dependent Riccati Equation-based (SDRE) control algorithm is applied [25–27]. In [28,29] a comparative study between SDRE and LQR is presented, and SDRE showed its advantages considering fuel consumption, rendezvous time, and trajectory accuracy. The SDRE-based algorithms are used to address various problems such as the position and attitude control of a single spacecraft [30] or relative motion control in satellite formation flying [29]. For the problem of space debris object capturing the kinematic coupling effect must be taken into account when the relative motion of not centers of mass of two bodies but motion between two defined body-fixed points is considered as in [31]. The paper [32] studies the application of the SDRE-based control for this type of relative motion equations. The main contribution of the current paper is in the study of the influence of the control system parameters on the performance of the SDRE-based control algorithm on the relative motion during the capturing, taking into account the reaction wheels saturation and thrusters misalignment.

This paper considers a spacecraft with thrusters installed on board to control the center of mass motion and it is equipped with reaction wheels for the attitude control [33]. The motion of a non-cooperative object relative to the spacecraft is considered known for the sake of simplicity. In real cases the target motion is estimated by relative motion determination system with some errors and uncertainties. The purpose of the work is to develop an algorithm for controlling both the center of mass motion and the angular motion to achieve the required relative position and attitude of the spacecraft relative to a non-cooperative object, which is necessary for the capturing. It is assumed that the object of space debris has an axis of dynamical symmetry and rotates freely under the influence of a gravitational torque and its tumbling motion is similar to nutation. To capture space debris it is necessary to match a desired point on the spacecraft reference frame with a point on the object surface [31,34]. As a capturing system can be considered a robotic arm capable of catching on an element of the space debris body.

SDRE control algorithm to achieve relative motion along a desired trajectory is developed in the paper [35]. The SDRE method requires linearization of the equations of motion in the vicinity of the current state [27], and the optimal controller coefficients are calculated as a result of solving the Riccati equation at each control cycle [36]. Nonlinear coupled translational and rotational motion equations of the spacecraft relative to the object are used. It is assumed that the value of the thrust is limited and discrete, and the thrust vector has a misalignment, which produces an additional disturbing torque and an external coupling effect [37]. Because of control limitations the successful capture of a space debris object is possible only in the region of acceptable values of the system parameters. The parameters include the initial conditions for the motion equations, the magnitude of the misalignment of the thrust, the value of the control constraints, the position of the capture point on the object, the parameters of the angular motion of the object. This paper is the continuation of authors pervious work [35], it considers a more generalized equations of motion and proposes a methodology for assessing the acceptable range of these parameters using a numerical study of the system motion.

## 2. Equations of Motions and Control Law

In this section a short introduction to equations of motion and control law is presented. The relative state vector consist of relative position and velocity between the centers of mass of a chaser (active satellite) and target (passive space debris) and relative attitude quaternion and angular velocity. Since for the capturing it is necessary to align two points on the surfaces of the satellites and in the object (the position of the capturing system of the chaser and capturing point of the target, see Figure 1), the relative rotational and translational motion equation are coupled.

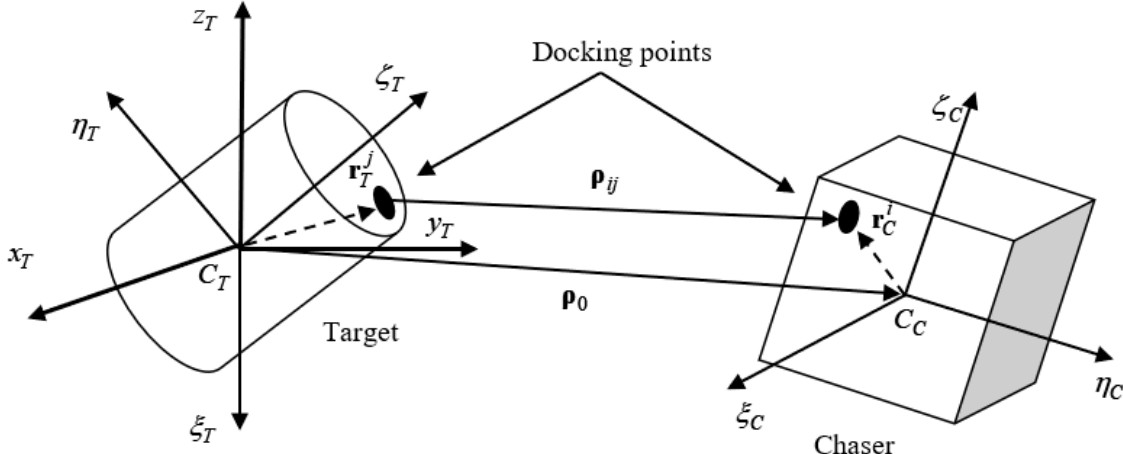

**Figure 1.** Reference frames of chaser satellite and target object.

### 2.1. Relative Rotational and Translational Equations of Motion

For modeling the dynamics of this problem the coupled sets of nonlinear rotational and translational relative equations of motion of spacecraft with respect to the non-cooperative object are used. The Equations (1) and (2) describe the changes of angular momentum of chaser $\mathbf{H}_C$ and the target $\mathbf{H}_T$ in inertial reference frame:

$$\left.\frac{d\mathbf{H}_C}{dt}\right|_I = \left.\frac{d\mathbf{H}_C}{dt}\right|_C + \boldsymbol{\omega}_C \times \mathbf{H}_C = \mathbf{N}_C + \mathbf{T}_C. \tag{1}$$

$$\left.\frac{d\mathbf{H}_T}{dt}\right|_I = \left.\frac{d\mathbf{H}_T}{dt}\right|_T + \boldsymbol{\omega}_T \times \mathbf{H}_T = \mathbf{N}_T \tag{2}$$

where $\square|_I$, $\square|_C$, and $\square|_T$ are derivative in the inertial, chaser-fixed, target-fixed reference frame respectively, $\boldsymbol{\omega}_C$, $\boldsymbol{\omega}_T$ are chaser and target angular velocities, $\mathbf{N}_C$ and $\mathbf{N}_T$ are external torques acting on the spacecraft and debris respectively, and $\mathbf{T}_C$ is the control torque. The chaser satellite is equipped with reaction wheels, so its angular momentum is calculated as follows:

$$\begin{aligned} \mathbf{H}_C &= \mathbf{I}_C\boldsymbol{\omega}_C + \mathbf{h}_{WC} \\ \mathbf{H}_t &= \mathbf{I}_T\boldsymbol{\omega}_T \end{aligned} \tag{3}$$

where $\mathbf{h}_{WC}$ is the angular momentum produced by reaction wheels, $\mathbf{I}_C$ and $\mathbf{I}_T$ are chaser and debris inertia tensor. It is assumed that the chaser satellite is equipped with three reaction wheels, the angular momentum of each wheel is aligned with principal axis of the satellite. All the variables in Equations (1) and (2) are expressed in the corresponding body-fixed reference frame.

Since the target is passive and the chaser is equipped with control system, the relative rotational equation of motion should be expressed in target-fixed reference frame. By subtracting (2) from (1) and

taking into account the transition matrix $\mathbf{D}(\mathbf{q})$ between the chaser reference frame to target reference frame the following relative angular motion equation is obtained:

$$\mathbf{I}_T\dot{\boldsymbol{\omega}}^T = \mathbf{I}_T\mathbf{D}(\mathbf{q})\mathbf{I}_C^{-1}[-\mathbf{D}(\mathbf{q})^{-1}(\boldsymbol{\omega}^T + \boldsymbol{\omega}_T^T) \times \mathbf{I}_C\mathbf{D}(\mathbf{q})^{-1}(\boldsymbol{\omega}^T + \boldsymbol{\omega}_T^T)$$
$$-\mathbf{D}(\mathbf{q})^{-1}(\boldsymbol{\omega}^T + \boldsymbol{\omega}_T^T)\mathbf{h}_{WC} - \dot{\mathbf{h}}_{WC} + \mathbf{T}_C + \mathbf{N}_C] - \mathbf{I}_T\boldsymbol{\omega}_T^T \times \boldsymbol{\omega}^T + [\boldsymbol{\omega}_T^T \times \mathbf{I}_T\boldsymbol{\omega}_T^T] \tag{4}$$

Here $\boldsymbol{\omega}^T = \mathbf{D}(\mathbf{q})\boldsymbol{\omega}_C{}^C - \boldsymbol{\omega}_T{}^T$ is the relative angular velocity which is expressed in the target's reference frame, superscript defines the reference frame where the value is expressed, "$C$" stands for chaser and "$T$" stands for non-cooperative target. The direction cosine matrix of rotation can be expressed using quaternion $\mathbf{q}$ that defines the transition from chaser to target reference frame.

The external torque $\mathbf{N}_T$ acting on the target includes only gravitational torque, but $\mathbf{N}_C$ also includes the torque $\mathbf{N}_{th}$ caused by thruster vector misalignment from the chaser center of mass.

$$\mathbf{N}_{th} = \mathbf{r}_{th} \times \mathbf{F}_{th}$$

where $\mathbf{F}_{th}$ is a thrust vector of the thruster, $\mathbf{r}_{th}$ is the radius vector from center of mass to the center of thrust application.

The kinematical equation can be written for quaternions as follow:

$$\dot{\mathbf{q}} = \frac{1}{2}\mathbf{Q}(\mathbf{q})\boldsymbol{\omega}^T \tag{5}$$

In (5) $\mathbf{Q}$ is given as:

$$\mathbf{Q} = \begin{bmatrix} q_4 & -q_3 & q_2 \\ q_3 & q_4 & -q_1 \\ -q_2 & q_1 & q_4 \\ -q_1 & -q_2 & -q_3 \end{bmatrix} \tag{6}$$

In Figure 1 the local-vertical-local-horizontal coordinate system of target is shown as $C_T x_T y_T z_T$. $C_T \xi_T \eta_T \zeta_T$ is the target body-fixed reference frame and $C_C \xi_C \eta_C \zeta_C$ is the chaser body-fixed reference frame of the chaser.

The general nonlinear relative equations of motion centers of mass of two objects are given as follows:

$$\ddot{x} - 2\omega_{OT}\dot{y} - \dot{\omega}_{OT}y - \omega_{OT}{}^2x = -\frac{\mu(r_T + x)}{\left[(r_T + x)^2 + y^2 + z^2\right]^{\frac{3}{2}}} + \frac{\mu}{r_T{}^2} + a_x \tag{7}$$

$$\ddot{y} + 2\omega_{OT}\dot{x} + \dot{\omega}_{OT}x - \omega_{OT}{}^2y = -\frac{\mu y}{\left[(r_T + x)^2 + y^2 + z^2\right]^{\frac{3}{2}}} + a_y \tag{8}$$

$$\ddot{z} = -\frac{\mu z}{\left[(r_T + x)^2 + y^2 + z^2\right]^{\frac{3}{2}}} + a_z \tag{9}$$

where $\boldsymbol{\rho}_0 = [x; y; z]^T$ is radius-vector of chaser center of mass in LHLV reference frame with target in the origin $C_T$, $r_T$ is the value of the radius-vector of the target center of mass in inertial reference frame; $\mathbf{a} = \begin{bmatrix} a_x & a_y & a_z \end{bmatrix}^T$ is the control acceleration, which is produced by thrusters; $\omega_{OT}$ is the orbital angular velocity of the target; $\mu$ is the Earth gravitational constant. In the case of near circular orbits and $r_T \gg \boldsymbol{\rho}_0$ these equations can be linearized and the well-known Clohessy-Wiltshire can be obtained [38]. To provide control acceleration it is assumed that the chaser satellite is equipped with six thrusters, a pair of thrusters with opposite directions for each principal axis of satellite.

To develop a control algorithm for capturing the target it is required to obtain the equations of relative motion of two points—the point of the chaser satellite where the capturing mechanism is placed, and the point of the target object for which the object can be captured. The coupling effect

between translational and attitude motion should be taken into account. Consider two arbitrary points of the target $\mathbf{r}_T^j$ and of the chaser $\mathbf{r}_C^i$ (see Figure 1). For these two points the relative motion equations can be derived from the following relation for the relative vector $\boldsymbol{\rho}_{ij}$ between these points:

$$\boldsymbol{\rho}_{ij} = \boldsymbol{\rho}_0 + \mathbf{r}_C^i - \mathbf{r}_T^j \tag{10}$$

All of the vectors are expressed in LVLH reference frame. Calculating the first and then second derivative the following equations can be obtained:

$$\dot{\boldsymbol{\rho}}_{ij} = \dot{\boldsymbol{\rho}}_0 + \boldsymbol{\omega} \times \mathbf{r}_C^i \tag{11}$$

$$\ddot{\boldsymbol{\rho}}_{ij} = \ddot{\boldsymbol{\rho}}_0 + \dot{\boldsymbol{\omega}} \times \mathbf{r}_C^i + \boldsymbol{\omega} \times (\boldsymbol{\omega} \times \mathbf{r}_C^i) \tag{12}$$

Taking into account that $\boldsymbol{\rho}_{ij} = \begin{bmatrix} x_{ij} \ y_{ij} \ z_{ij} \end{bmatrix}^T$ Equation (12) can be rewritten in the following form:

$$\begin{aligned}
\ddot{x}_{ij} - 2\omega_{OT}\dot{y}_{ij} - \dot{\omega}_{OT}y_{ij} - 3\omega_{OT}{}^2 x_{ij} &= a_x + K_1 \\
\ddot{y}_{ij} + 2\omega_{OT}\dot{x}_{ij} + \dot{\omega}_{OT}x_{ij} &= a_y + K_2 \\
\ddot{z}_{ij} + \omega_{OT}^2 z_{ij} &= a_z + K_3
\end{aligned} \tag{13}$$

where $\mathbf{K} = [K_1 \ K_2 \ K_3]^T$ are terms which consist of $\boldsymbol{\omega}$ and express the coupling between (13) and (4):

$$\begin{aligned}
K_1 &= \left[ \omega_y\left(\omega_x y_C^i - \omega_y x_C^i\right) - \omega_z\left(\omega_z x_C^i - \omega_x z_C^i\right) \right] + \dot{\omega}_y z_C^i - \dot{\omega}_z y_C^i \\
&\quad + 2\omega_{OT}\left(-\omega_z x_C^i + \omega_x z_C^i\right) + \dot{\omega}_{OT}\left(-y_C^i + y_T^j\right) + 3\omega_{OT}^2\left(-x_C^i + x_T^j\right) + \frac{\mu}{r_T^2} - \frac{\mu r_T}{r_C^3}
\end{aligned} \tag{14}$$

$$\begin{aligned}
K_2 &= \left[ \omega_z\left(\omega_y z_C^i - \omega_z y_C^i\right) - \omega_x\left(\omega_x y_C^i - \omega_y x_C^i\right) \right] + \dot{\omega}_z x_C^i - \dot{\omega}_x z_C^i \\
&\quad + 2\omega_{OT}\left(\omega_y z_C^i + \omega_z y_C^i\right) + \dot{\omega}_{OT}\left(-x_C^i + x_T^j\right) + \omega_{OT}^2\left(-y_C^i + y_T^j\right)
\end{aligned} \tag{15}$$

$$K_3 = \left[ \omega_x\left(\omega_z x_C^i - \omega_x z_C^i\right) + \omega_y\left(\omega_y z_C^i - \omega_z y_C^i\right) \right] + \dot{\omega}_x y_C^i - \dot{\omega}_y x_C^i + \omega_{OT}\left(z_C^i - z_T^j\right) \tag{16}$$

where $\mathbf{r}_T^j = [x_T^j \ y_T^j \ z_T^j]^T$ is the radius-vector of the capturing point of the target and $\mathbf{r}_C^i = [x_C^i \ y_C^i \ z_C^i]^T$ is the radius-vector of the capturing system of the chaser.

Note that these coupled equations are expressed in LVLH placed in the target. However, for the onboard algorithms is it necessary to develop the equations in body-fixed reference frame. This issue is addressed in the next chapter.

## 2.2. Modified Coupled Translational Equations of Motion

As mentioned above $\ddot{\boldsymbol{\rho}}_{ij}$ is expressed in LVLH and it is required to be expressed in body reference frame of the target. The relation between radius-vector between the centers of mass $\boldsymbol{\rho}_0$ in the LVLH reference frame and its value $\widetilde{\boldsymbol{\rho}}_0$ in target body-fixed reference frame is as follows:

$$\widetilde{\boldsymbol{\rho}}_0 = \mathbf{A}(\mathbf{q}_T)\boldsymbol{\rho}_0 \tag{17}$$

where $\mathbf{q}_T$ is quaternion of the target reference frame relative to LVLH, $\mathbf{A}(\mathbf{q}_T)$ is the direction cosine matrix. The kinematical relations for direction cosine matrix can be written as:

$$\dot{\mathbf{A}} = \boldsymbol{\Omega}\mathbf{A}, \tag{18}$$

where $\boldsymbol{\Omega}$ is skew-symmetric matrix filled using target angular velocity relative to LVLH. Taking time derivative from (17) and substituting (18) one can obtain the following:

$$\dot{\widetilde{\boldsymbol{\rho}}}_0 = \boldsymbol{\Omega}\widetilde{\boldsymbol{\rho}}_0 + \mathbf{A}\dot{\boldsymbol{\rho}}_0. \tag{19}$$

From this equation we get the following:

$$\dot{\boldsymbol{\rho}}_0 = \mathbf{A}^T\left(\dot{\widetilde{\boldsymbol{\rho}}}_0 - \boldsymbol{\Omega}\widetilde{\boldsymbol{\rho}}_0\right) \tag{20}$$

Taking derivative from (19) acceleration relationship can be found

$$\ddot{\widetilde{\boldsymbol{\rho}}}_0 = \dot{\boldsymbol{\Omega}}\widetilde{\boldsymbol{\rho}}_0 - \boldsymbol{\Omega}^2\widetilde{\boldsymbol{\rho}}_0 + 2\boldsymbol{\Omega}\dot{\widetilde{\boldsymbol{\rho}}}_0 + \mathbf{A}\ddot{\boldsymbol{\rho}}_0 \tag{21}$$

On the other hand we can rewrite Clohessy-Whiltshire equations [33] in vector-matrix form as:

$$\ddot{\boldsymbol{\rho}}_0 + \mathbf{M}\dot{\boldsymbol{\rho}}_0 + \mathbf{N}\boldsymbol{\rho}_0 = \mathbf{a} + \mathbf{K} \tag{22}$$

where

$$\mathbf{N} = \begin{bmatrix} -3\omega_{OT}{}^2 & -\dot{\omega}_{OT} & 0 \\ \dot{\omega}_{OT} & 0 & 0 \\ 0 & 0 & \omega_{OT}^2 \end{bmatrix}$$
$$\mathbf{M} = \begin{bmatrix} 0 & -2\omega_{OT} & 0 \\ 2\omega_{OT} & 0 & 0 \\ 0 & 0 & 0 \end{bmatrix} \tag{23}$$

The Equation (22) is written in LVLH. Rewrite both sides of the equation is in target body-fixed reference frame:

$$\mathbf{A}^T\left[\ddot{\widetilde{\boldsymbol{\rho}}}_0 - \dot{\boldsymbol{\Omega}}\widetilde{\boldsymbol{\rho}}_0 + \boldsymbol{\Omega}^2\widetilde{\boldsymbol{\rho}}_0 - 2\boldsymbol{\Omega}\dot{\widetilde{\boldsymbol{\rho}}}_0\right] + \mathbf{M}\mathbf{A}^T\left(\dot{\widetilde{\boldsymbol{\rho}}}_0 - \boldsymbol{\Omega}\widetilde{\boldsymbol{\rho}}_0\right) + \mathbf{N}\mathbf{A}^T\widetilde{\boldsymbol{\rho}}_0 = \mathbf{A}^T\widetilde{\mathbf{a}} + \mathbf{A}^T\widetilde{\mathbf{K}} \tag{24}$$

where $\widetilde{\mathbf{K}} = \mathbf{A}\mathbf{K}$, $\widetilde{\mathbf{a}} = \mathbf{A}\mathbf{a}$. From this equation one can rewrite it as follows:

$$\ddot{\widetilde{\boldsymbol{\rho}}}_0 = \widetilde{\mathbf{a}} + \widetilde{\mathbf{K}} + \boldsymbol{\beta}\dot{\widetilde{\boldsymbol{\rho}}}_0 + \boldsymbol{\gamma}\widetilde{\boldsymbol{\rho}}_0 \tag{25}$$

where

$$\boldsymbol{\beta} = -\mathbf{A}\left[\mathbf{M}\mathbf{A}^T - 2\mathbf{A}^T\boldsymbol{\Omega} - \mathbf{A}\dot{\boldsymbol{\Omega}}\right]$$
$$\boldsymbol{\gamma} = -\mathbf{A}\left[\mathbf{N}\mathbf{A} - \mathbf{M}\mathbf{A}^T\boldsymbol{\Omega} + \mathbf{A}\boldsymbol{\Omega}^2\right] \tag{26}$$

So, the Equation (25) are expressed in body-fixed reference frame of target. Now it is possible to derive coupled equations of motion between two points expressed in the target body frame. Let the points of the target $\mathbf{r}_T^j$ and of the chaser $\mathbf{r}_C^i$ are written in target reference frame. Then, using (10) and substituting $\widetilde{\boldsymbol{\rho}}_0$, $\dot{\widetilde{\boldsymbol{\rho}}}_0$, $\ddot{\widetilde{\boldsymbol{\rho}}}_0$ instead of $\boldsymbol{\rho}_0$, $\dot{\boldsymbol{\rho}}_0$, $\ddot{\boldsymbol{\rho}}_0$ in (12) the following equation is obtained:

$$\ddot{\widetilde{\boldsymbol{\rho}}}_{ij} = \boldsymbol{\beta}\dot{\widetilde{\boldsymbol{\rho}}}_{ij} + \boldsymbol{\gamma}\widetilde{\boldsymbol{\rho}}_{ij} + \left([\boldsymbol{\omega}^T]_x^2 - \boldsymbol{\beta}[\boldsymbol{\omega}^T]_x + [\dot{\boldsymbol{\omega}}^T]_x\right)\mathbf{r}_C^i + \boldsymbol{\gamma}\mathbf{r}_T^j + \widetilde{\mathbf{a}} + \mathbf{K} \tag{27}$$

These equations of the relative motion of two arbitrary points of the target and the chaser is used for control algorithm development.

### 2.3. SDRE Control Algorithm

The common nonlinear motion equations of the considered system is as follows:

$$\dot{\mathbf{x}} = \mathbf{f}(\mathbf{x}(t)) + \mathbf{g}(\mathbf{x}(t), \mathbf{u}(t)) \tag{28}$$

Here $\mathbf{x} \in \mathbb{R}^n$ is the state vector; $\mathbf{u} \in \mathbb{R}^m$ is the control vector, $\mathbf{f}(\ ), \mathbf{g}(\ )$ are nonlinear smooth functions; for establishing the SDRE control algorithm for the dynamical system of type (28) the functional $J$ of type (29) is considered:

$$J = \frac{1}{2} \int_0^{t_f} \left[ \mathbf{x}(t)^T \mathbf{Q} \, \mathbf{x}(t) + \mathbf{u}(t)^T \mathbf{R} \, \mathbf{u}(t) \right] dt \tag{29}$$

where $\mathbf{Q}, \mathbf{R}$ are positive definite weighting matrices. In (29) finite horizon $t_f$ is considered. Here the point $\mathbf{x} = 0$ is assumed to be equilibrium of the system. In (29) the minimization of the infinite horizon cost function is required. SDRE method requires the linearization of the equation of motion in a neighborhood of the equilibrium. The optimal coefficients of the regulator are calculated as a result of solving Riccati equation at each time step. In this paper $\mathbf{Q}, \mathbf{R}$ are considered as constant matrices. Next step is to linearize the nonlinear system. After linearization, the dynamical system has the following form:

$$\dot{\mathbf{x}} = \mathbf{A}(\mathbf{x})\mathbf{x} + \mathbf{B}(\mathbf{x})\mathbf{u} \tag{30}$$

where $\mathbf{A}(\mathbf{x})$ is the matrix of dynamic and $\mathbf{B}(\mathbf{x})$ is the control matrix. Similar to linear quadratic regulator a corresponding algebraic Riccati equation can be derived for nonlinear quadratic regulator. After forming the Hamilton function and using the maximum principle of Pontryagin [36], necessary conditions for optimality can be applied to obtain the optimal control law as (31)

$$\mathbf{u}(\mathbf{x}) = \mathbf{R}^{-1}\mathbf{B}^T(\mathbf{x})\mathbf{P}(\mathbf{x})\mathbf{x} \tag{31}$$

The control function is similar to LQR, but unlike LQR here the coefficients are functions of state vector. The matrix $\mathbf{P}(\mathbf{x})$ is unique, symmetric, and positive-definite and can be achieved by solving following Riccati equation:

$$\mathbf{P}(\mathbf{x})\mathbf{A}(\mathbf{x}) + \mathbf{A}^T(\mathbf{x})\mathbf{P}(\mathbf{x}) - \mathbf{P}(\mathbf{x})\mathbf{B}(\mathbf{x})\mathbf{R}^{-1}(\mathbf{x})\mathbf{B}^T(\mathbf{x})\mathbf{P}(\mathbf{x}) + \mathbf{Q}(\mathbf{x}) = 0 \tag{32}$$

The closed loop system using this semi-optimal control (31):

$$\dot{\mathbf{x}} = \left( \mathbf{A} - \mathbf{B}(\mathbf{x})\mathbf{R}^{-1}\mathbf{B}^T(\mathbf{x})\mathbf{P}(\mathbf{x}) \right)\mathbf{x} \tag{33}$$

The Riccati equation can be solved analytically using Hamiltonian matrix [39,40].

## 2.4. Control Application to the Problem of Capturing

In this paper the state vector $\mathbf{x}(t)$ consisting of 12 components is considered. The state vector is composed of three vector-part quaternion components of relative attitude between the target and the chaser $[q_1, q_2, q_3]^T$, the angular velocity vector $[\omega_x, \omega_y, \omega_z]^T$, relative translational position, and velocity of docking points $\left[ \widetilde{x}_{ij}, \widetilde{y}_{ij}, \widetilde{z}_{ij}, \dot{\widetilde{x}}_{ij}, \dot{\widetilde{y}}_{ij}, \dot{\widetilde{z}}_{ij} \right]^T$. The control vector $\mathbf{u}(t)$ contains six elements to provide a full feedback control for the coupled motion. The three first elements of vector $\mathbf{u}(t)$ are reaction wheels control vector $\mathbf{h}_{WC}$ and thrusters force vector $\mathbf{F}$. The matrix of dynamic $\mathbf{A}(\mathbf{x})$ and control matrix $\mathbf{B}(\mathbf{x})$ are as follows:

$$\mathbf{A} = \begin{bmatrix} \begin{bmatrix} \left[\mathbf{D}^T(q)\boldsymbol{\omega}^T\right]_x & q_4\mathbf{D}^T \\ 0_{3\times3} & \mathbf{Ro} \end{bmatrix} & 0_{6\times6} \\ 0_{6\times6} & \begin{bmatrix} 0_{3\times3} & \mathbf{I}_{3\times3} \\ \gamma & \beta \end{bmatrix} \end{bmatrix}. \tag{34}$$

$$\mathbf{B} = \begin{bmatrix} 0_{3\times3} & 0_{3\times3} \\ -\mathbf{DI}_C^{-1} & 0_{3\times3} \\ 0_{3\times3} & 0_{3\times3} \\ 0_{3\times3} & \mathbf{I}_{3\times3}/m \end{bmatrix}, \tag{35}$$

where

$$\mathbf{Ro} = -\left[\mathbf{DI}_C^{-1}\mathbf{D}^T\boldsymbol{\omega}^T\right]\mathbf{I}_C\mathbf{D}^T + [\mathbf{h}_{WC}]_x\mathbf{DI}_C^{-1}\mathbf{D}^T - \left[\mathbf{DI}_C^{-1}\mathbf{D}^T\boldsymbol{\omega}_T^T\right]_x\mathbf{I}_C\mathbf{D}^T - \left[\boldsymbol{\omega}_T^T\right]_x + \left[\mathbf{I}_C\mathbf{D}^T\boldsymbol{\omega}_T^T\right]_x\mathbf{DI}_C^{-1}\mathbf{D}^T \tag{36}$$

Since the control is aimed to coincide attitude of the target and the chaser, then the docking points of the target $\mathbf{r}_T^j$ and of the chaser $\mathbf{r}_C^i$ should be chosen in order not to avoid the collision of them. From the mathematical point of view it means the constraint that can be written as scalar product of these two vectors must be negative, i.e., $\left(\mathbf{r}_T^j, \mathbf{r}_C^i\right) < 0$. In the case when the capturing point of the target is not satisfied then we can apply an additional rotation to chaser with attitude matrix $\mathbf{S}$, it means that the point will be calculated as $\mathbf{Sr}_C^i$.

## 3. Numerical Study

Consider a demonstration of the proposed docking algorithm application. At the initial time the parameters of the target objects is set according to the Table 1. The chaser satellite is initially has no angular velocity with respect to the LVLH, but the target has defined initial rotation. The relative initial conditions for the attitude motion and for the translational motion is also presented in the right part of the Table 1.

**Table 1.** Parameters and initial condition of the modelling.

| Orbital Parameters | | Initial Conditions | |
|---|---|---|---|
| Altitude, km | 750 | $\mathbf{q}(t_0) = \mathbf{q}_T^T(t_0) = \mathbf{q}_C^C(t_0)$ | $[0,0,0,1]^T$ |
| Eccentricity | 0.03 | $\boldsymbol{\omega}(t_0) = \boldsymbol{\omega}_T^T(t_0)$, deg/s $\boldsymbol{\omega}_C^C(t_0)$, deg/s | $[10,-10,20]^T [0,0,0]^T$ |
| Inclination, deg | 70 | $\boldsymbol{\rho}(t_0) = [x_0,y_0,z_0]^T$, m | $[50,27,100]^T$ |
| Right ascension, deg | 50 | $\dot{\boldsymbol{\rho}}(t_0) = \dot{\mathbf{r}}_0$, m/s | $[0,-2,0]^T$ |
| Argument of perigee, deg | 80 | $\mathbf{r}_C^i$, m | $[-1, 1, 0]^T$ |
| Initial true anomaly | 0 | $\mathbf{r}_T^j$, m | $[1,0,1]^T$ |

In this example the weight matrix $\mathbf{Q}$ is set to identity matrix and matrix $\mathbf{R}$ is block matrix of the following structure:

$$\mathbf{R} = \begin{bmatrix} \mathbf{R}_{rot} & 0_{3x3} \\ 0_{3x3} & \mathbf{R}_{tran} \end{bmatrix} \tag{37}$$

The matrix $\mathbf{R}_{rot}$ equals to $160 \cdot \mathbf{I}_{6x6}$ and $\mathbf{R}_{tran} = 160 \cdot \mathbf{I}_{6x6}$. These parameters are chosen using trial and error approach in order not to meet reaction wheels saturation. It is assumed in the paper that $\mathbf{R}_{tran}$ is fixed, but the influence of $\mathbf{R}_{rot}$ on the algorithm is studied. The thruster misalignment in this example is set to 3.5 mm. In this example the inertia tensor of the target is set to identity matrix, and the inertia tensor of the chaser is $2 \cdot \mathbf{I}_{3x3}$. The maximal reaction wheel kinematic momentum is chosen as 1 Nms, which corresponds to reaction wheels mini-wheels produced by Honeywell company [41].

For the docking, it is necessary to control motion in a way that relative distance and velocity (both of rotational and translational motion) converge to zero. These relative position and velocity are shown correspondingly in Figures 2 and 3. This relative translational motion is shown in Figure 4.

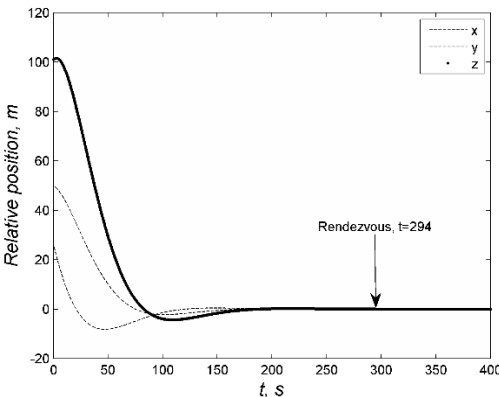

**Figure 2.** Relative position with respect to the target during the capturing maneuver.

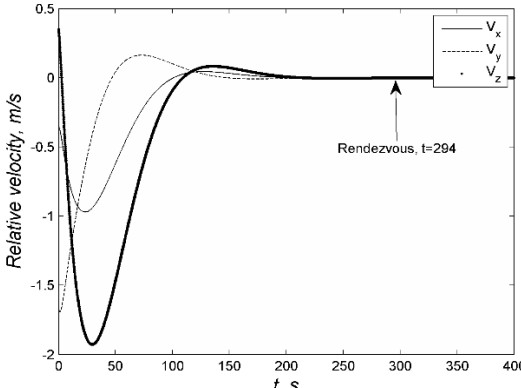

**Figure 3.** Relative velocity with respect to the target during the capturing maneuver.

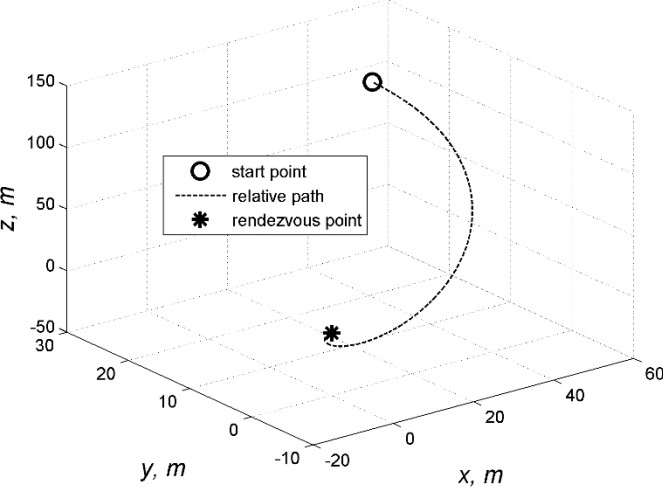

**Figure 4.** Relative motion of satellite to target.

The attitude of the chaser and angular velocity with respect to the target is shown in Figures 5 and 6. At the time of about 135 s quaternion parameters achieve values less than $10^{-5}$ (about $10^{-3}$ deg of the deviation) and it is assumed as the required accuracy for attitude tracking. At the same time the position errors are quite large and the more control effort from thrusters are required to obtain rendezvous. When the relative distance becomes less than 1 cm the docking is assumed to be accomplished. This time is longer and is 294 which is shown in Figures 2 and 3.

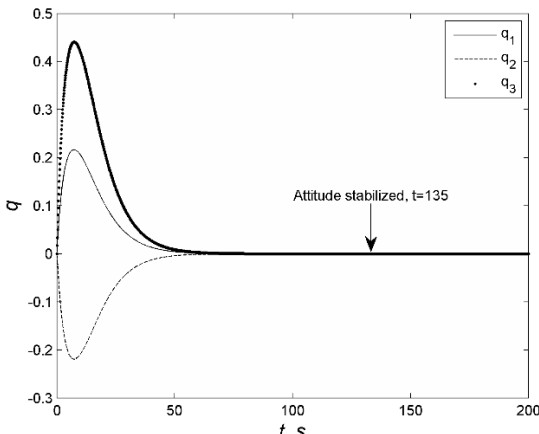

**Figure 5.** Relative quaternion with respect to the target.

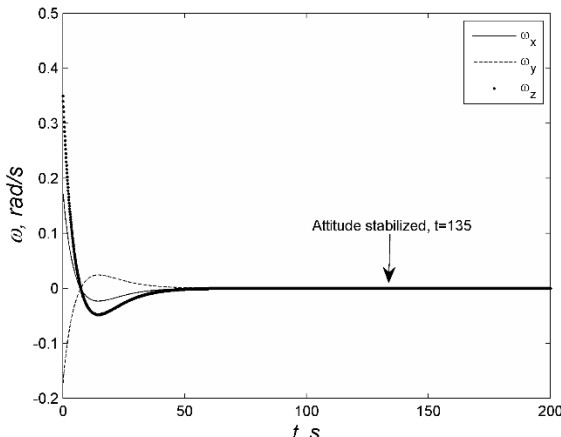

**Figure 6.** Relative angular velocity with respect to the target.

Figure 7 shows the discrete thrust values that are used to control translational motion. The value of the discreet thrust is 0.5 N. Figure 8 shows angular momentum of reaction wheels that achieved their final constant values to hold to tracking attitude. Because of thruster misalignment and in order to track the target attitude the reaction wheels angular momentum is increased but still do not exceed the maximal values of 1 Nms.

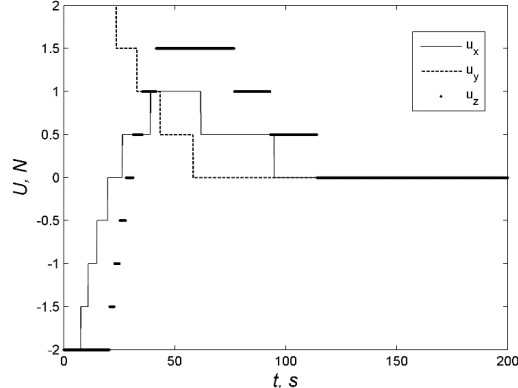

**Figure 7.** Thrust vector of satellite.

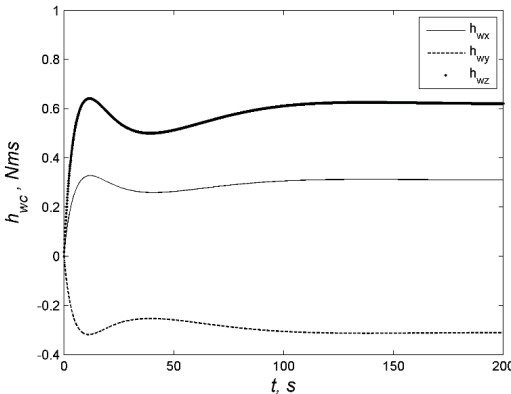

**Figure 8.** Angular momentum of reaction wheels.

Reaction wheels have operational limitation, exceeding these limits will lead to saturation, which is caused not only by the thruster misalignment but also by improper choices of control algorithm parameters such as weighting matrix $\mathbf{R}_{rot}$. The inertia tensor of the target and its rotational velocity can cause saturation as well. Here the task is to determine the maximum values of angular velocity of the target, the misalignment of thrust vector, the weighting matrix at which the capturing maneuver will be successful, i.e., the required attitude and points positions deviation will be achieved. After numerical simulation it will be possible to determine the operational range of reaction wheels under critical variables to avoid the wheels saturation.

Consider a target with inertia tensor that is not identity matrix, but a inertia tensor of body with dynamical symmetry as the following:

$$\mathbf{I}_T = diag(J, J, kJ) \tag{38}$$

where $J$ is inertia moment along principal axis, $k$ is the ration coefficient for the inertia moment along axis of dynamical symmetry. According to physical constraint of the inertia moments $k > 0.5$. It is assumed that the inertia tensor of the target is known. In practical cases it is estimated at the stage of space debris observation, for example using image processing technique [42]. The target with dynamical symmetry is chosen for consideration because in that case it is easier to study the influence of the inertial moment of the target on control algorithm performance. Moreover, a lot of debris such as upper stages of rockets and some inactive satellites are close to body with dynamical symmetry. Taking all the parameters from the previous simulation example except the target inertia tensor, the maximal value of the reaction wheels momentum is calculated for changing ratio coefficient $k$. The results are shown in Figure 9. It can be seen that the lowest maximal angular momentum is close to the case when $k = 1$ that correspond to spherical body. In the case when the body has a dynamical symmetry the more the ratio coefficient of matrix of inertia, the higher the maximal angular momentum of the reaction wheels. It can be explained by the increasing influence of the nutation motion of the body, the capturing point of the target position is harder to track by the reaction wheels of the chaser. To further study the influence of nutation motion on the control algorithm performance, the initial angular velocity is also varied. For the random ratio $k$ from the interval $k \in [0.5, 3]$ and for the random initial angular velocity of the target (each component value is chosen from interval $[0, 24]$ deg/s) after simulation the maximum value of the reaction wheels momentum is obtained. Figure 10 shows these points for each simulation and corresponding values of maximum angular momentum which is required to be provided by reaction wheels. Almost all of the maximal values of the reaction wheels are less than 1 Nms, that is assumed to be the upper limit. However, in the case the angular velocity is higher that 24 deg/s the wheels saturation often occurs that causes the inappropriate errors for the capturing.

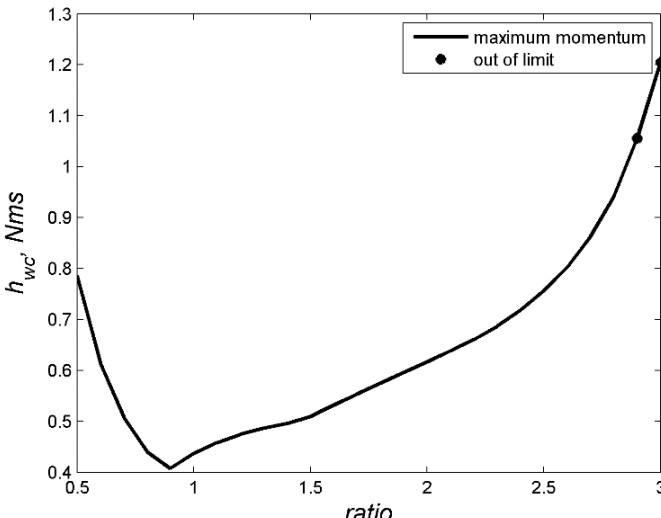

**Figure 9.** Maximum angular momentum with respect to inertial moment of the target.

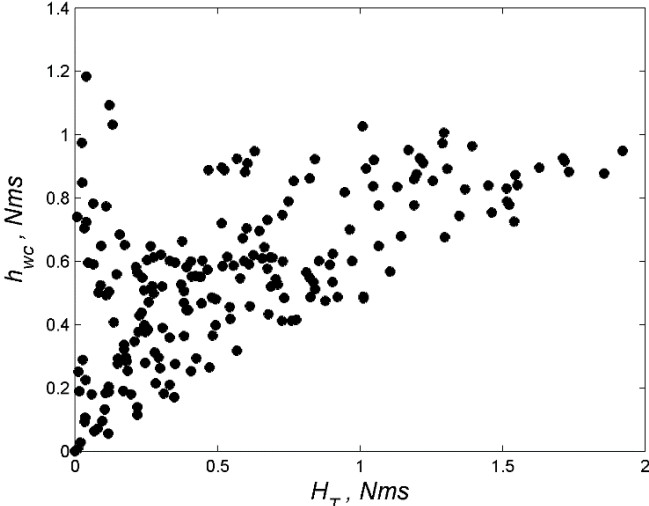

**Figure 10.** Maximum angular momentum of reaction wheels with respect to the target angular moment value.

The reaction wheels saturation can be also caused by thruster misalignment. To study its effect on the capturing the random misalignment $l \in [0, 0.01]$ m is added to simulations additionally to the varying parameters $k$ and angular velocity of the target. Moreover, the control parameter weighting matrix $\mathbf{R}_{rot}$ affects the required angular momentum of reaction wheel, consequently the changing elements of $\mathbf{R}_{rot} \in [120, 200]$ are added as well to the simulation alongside other varying parameters. The results of multiple simulations are presented in the Figure 11 in the parameters ranges, where bubbles mean that there is no saturation and the capturing is successful, and the stars are for capturing the failure due to reaction wheels saturation. The smaller the bubbles the less the maximal angular momentum of the reactions wheels.

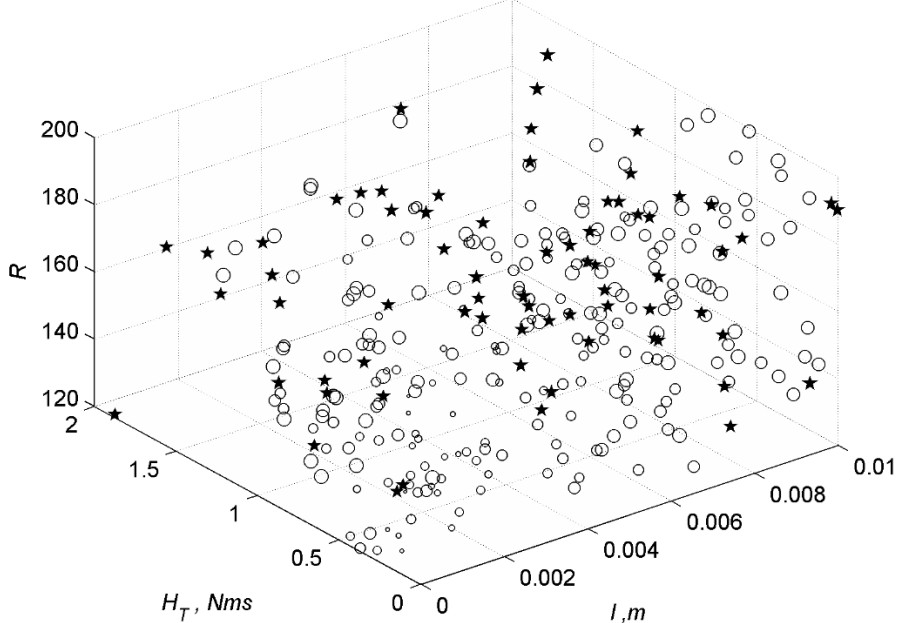

**Figure 11.** Random points and range of saturation of reaction wheels (bubbles means that there is no saturation and the capturing is successful, and the stars are for the capturing failure due to reaction wheels saturation).

To interpret Figure 11 it will be helpful to present the box diagram of influential parameters. Figure 12 shows the dependence of maximum angular momentum of reaction wheels to the misalignment of thrusters from the center of mass. The 50% of the simulation results are inside the rectangular and each outside interval contains 25% of results, the mean value is depicted as horizontal line in the box. Figure 12 demonstrates a gradual increase in angular momentum which intuitively could be predicted. Starting from 8 mm of thruster misalignment some of the simulations results exceed the limit of the angular momentum. Figure 13 shows the effect of angular momentum of target on maximum angular momentum of reaction wheels. In the given scale the increment seems that the lower values are exponentially converging to the value of 0.9 Nms. Using Figure 11 with a fixed $H_T$ and $l$ it is possible to choose the optimum value for control weighting matrix $\mathbf{R}_{rot}$.

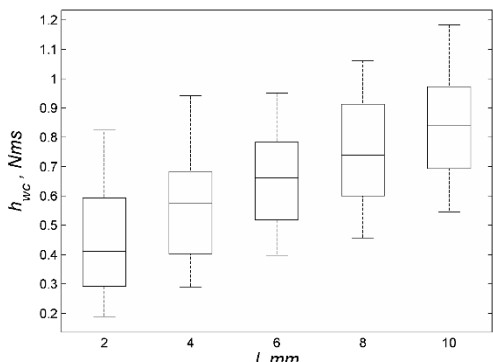

**Figure 12.** Box diagram of eccentricity of thrusters vector and angular momentum of reaction wheels.

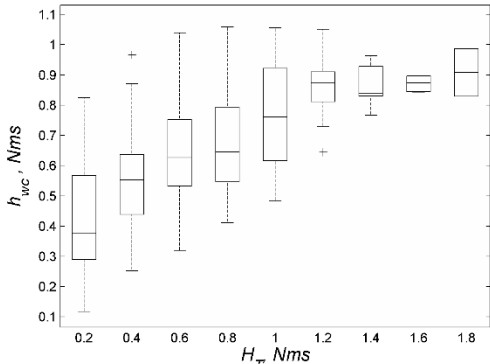

**Figure 13.** Box diagram of target's angular momentum vector and angular momentum of reaction wheels.

Thus, the developed relative coupled motion equations and SDRE-based control algorithm allow to study the influence of the most crucial parameters influence on the capturing performance. The multiple numerical study showed that for the presented case with defined parameters the successful capturing is possible when the thruster misalignment does not exceed 8 mm, the control weighting matrix $\mathbf{R}_{rot}$ elements are inside the interval $[120, 200]$. Moreover, the successful capturing is limited to the target angular velocity of about 24 deg/s for each component and inertia tensor ratio $k \in (0.5, 2.8]$. It should be noted that these results will be different for different initial relative vector and for other limit of the reaction wheels momentum but using the same methodology it is possible to obtain the allowable range for the successful capturing.

## 4. Conclusions

In this work an active space debris removal approach is considered. An algorithm for capturing a non-cooperative target in LEO orbit is presented. Moreover, a method to explore the boundaries and limitation of this suggested algorithm is developed. This method is used to determine the range of the acceptability of the algorithm with respect to the parameters such as targets angular momentum, and misalignment of the thrusters to avoid saturation of reaction wheels. For a debris with given moment on inertia and angular velocity this method allows to predict the possibility of tracking the target and capturing it. The multiple numerical study showed that for the presented case with defined parameters the successful capturing is possible when the thruster misalignment does not exceed 8 mm, the control weighting matrix $\mathbf{R}_{rot}$ elements are inside the interval $[120, 200]$. Moreover, the successful capturing is limited to the target angular velocity of about 24 deg/s for each component and inertia tensor ratio $k \in (0.5, 2.8]$. It should be noted that these results will be different for different initial relative vector and for other limit of the reaction wheels momentum but using the same methodology it is possible to obtain the allowable range for the successful capturing.

**Author Contributions:** M.A. conducted numerical study, D.I. proposed the main theory. All authors have read and agreed to the published version of the manuscript.

**Funding:** The work is supported by the Russian Foundation of Basic Research, grants NO 18-31-20014, 20-31-90072.

**Conflicts of Interest:** The authors declare no conflict of interest regarding the publication of this paper.

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
