# Peer review of "Influence of Satellite Motion Control System Parameters on Performance of Space Debris Capturing"

_aerospace, doi:10.3390/aerospace7110160_

Round 1

Reviewer 1 Report

The paper shows an interesting (but not new) application of the SDRE control to the problem of rendezvous to a tumbling body. The paper is well structured but needs some corrections with some sentences (see comments below), and it is recommended a revision of the introduction and abstract by a english mother tongue reviewer. The methodology is well described and results well presented, but the authors should highlight more the innovative content with respect to existing papers in literature. A minor revision is suggested, especially for solving the following points:

  • Page 1, Line 15. Please use range instead of area.
  • A literature review about the applications of SDRE techniques is missing in the introduction. This will also be useful for highlighting the innovative contributions of this paper against the already existing SDRE methods in the literature. The authors can refer to the exemplar works of:
  • T. Stansbery, J.R. Cloutier, "Position and Attitude Control of a Spacecraft Using the State-Dependent Riccati Equation Technique", Proceedings of the American Control Conference, Chicago (2000).
  • Felicetti, G.B. Palmerini, “A Comparison among Classical and SDRE Techniques in Formation Flying Orbital Control”, Proceedings of 2013 IEEE Aerospace Conference, Big Sky (2013)
  • Lee, H. Bang, “Coupled Position and Attitude Control of a Spacecraft in the Proximity of a Tumbling Target”, International academy of Astronautics Conference on Dynamics and Control of Space Systems, DyCoSS 2012, Porto.

The authors should also explain the main difference between this work and the previously published work:

  • Akhloumadi, M.; Ivanov, D. Satellite relative motion SDRE-based control for capturing a 380 noncooperative tumbling object. In Proceedings of the Proceedings of 9th International Conference Recent Advances in Space Technologies, RAST 2019

  • The introduction should stress more on the innovative content of the paper that, in my opinion, is represented by the results obtained with parametric analysis performed at the end of the numerical section rather than only the development and outline of the control algorithm itself (that probably is already existing in literature, see coment above).
  • Page 1, Line 36-37. The sentence “The reaction [… ] the object” is unclear and can mislead the reader.
  • Page 1, Line 40. Please simplify and correct the sentence. A possible wording can be: “The major part control algorithms for non-cooperative rendezvous and docking are optimal either in time or fuel”.
  • Page 2, Line 52. Please add “for the” and remove “system” so that the sentence reads as: “ with reaction wheels for the attitude control”
  • Page 2, Line 58. The utilization of “to coincide” in the sentence seems not appropriate.
  • Page 2, Line 66 and 69. What do the authors mean with “thrust vector has an eccentricity”? Please consider substituting eccentricity with “misalignment”.
  • Page 2, Line 77. I would suggest to substitute “area” with “range”. Please also change this in the abstract.
  • Page 5, Line 170, The Clohessy-Whiltshire equations appear for the first time in the article without any reference. Please add such a citation.
  • Page 7, Line 204, please substitute “Ricatti” with “Riccati”
  • Page 8, Line 246, Please explain how did you choose the coefficient 160 that is present in Rrot and Rtrans. In general, the actuators for position and attitude have different characteristics and even units. It appears very strange that these weights can have the same value. A specific study for selecting the best coefficients is necessary.
  • A comparative study between the SDRE and the LQR based on the linearized equations of motion is suggested to highlight the benefits of using such a methodology.
  • Page 11, Line 281, the sentence “In the case […] wheels” seems not well constructed. Please correct it.
  • Page 11, bottom part, change “parameters area” with “parameters ranges”
  • The conclusion section does not report the main findings obtained with the simulations and the performed parametric analysis. I would suggest the authors to include them.

Reviewer 2 Report

The paper presents the equations of relative motion between an uncontrolled rotating target and a fully controlled chaser satellites and a quasi-linearized control algorithm for the close-range rendezvous between them. The authors presents the full set of equations and the logic behind the control; a second part of the paper focuses on a numerical application of such method, with the evaluation of control parameters.

The mathematical formulation is well written and it is presented with enough detail. The numerical study indicates the quality of the proposed method. The authors introduce a series of assumptions that clearly simplify the application case. It is suggested to spend few words on the following assumptions that might greatly influence the real application of the proposed algorithm:

  • Row 53: the motion of the target is considered known. In a real-case application the target motion is usually not known and in case of in-situ measurements errors and uncertainties may arise. Please comment on that.
  • Row 57: please expand the sentence explaining the “axis of dynamical symmetry”. Generally uncontrolled bodies in space do not spin on one axis but tumble.
  • A saturation value of 1 Nms is really high for a space application. Do you have any reference to support the choice of this value?
  • Row 246, please explain the numerical value of 160 and how it was chosen. This might help in clarify the values reported in row 300.

Last, it is suggested to improve table 1 adding not only the relative motion parameters but also the chaser and the target initial conditions.

Reviewer 3 Report

Review of Influence of Satellite Motion Control System Parameters on Performance of Space Debris Capturing

This paper develops a SDRE based 6-DOF relative motion control for rendezvous with tumbling space debris object. The paper is well structured and easy to follow. I have the following comments:

  1. The authors state “The major part of the developed in the literature control algorithms for docking with non-cooperative are optimal either in time or fuel” However, there is a vast literature on 6-DOF relative motion control and the authors should extent their literature to reflect this. Some recent examples are:

Finite-time attitude set-point tracking for thrust-vectoring spacecraft rendezvous - Aerospace Science and Technology, 2020

Finite-time relative orbit-attitude tracking control for multi-spacecraft with collision avoidance and changing network topologies - Advances in Space Research, 2019

Extended-state-observer based event-triggered orbit-attitude tracking for low-thrust spacecraft - IEEE Transactions on Aerospace and Electronic, 2019

There are many more.

The author’s need to make much more effort to capture the state-of-the art in the field of 6-DOF relative motion control of spacecraft (which is essentially the problem addressed here).

  1. The author’s seem to have missed a critical paper related to the application of SDRE to 6-DOF relative motion control of spacecraft. The authors should carefully explain how the two control approaches differ in the introduction:

Application of SDRE technique to orbital and attitude control of spacecraft formation flying - Acta Astronautica, 2014

  1. More assumptions are required in the model i.e. you assume that the inertia of the target is known, how do you know the inertia of an uncooperative target? Is your control robust to uncertainties?
  2. The authors should include the actuator models of thrusters and reaction wheels and state the configuration of the thruster. How is the control allocated to each thruster or reaction wheel? Are the values feasible given the size of the spacecraft and the current technology. A comment on this would be useful.
  3. Why is a symmetric target assumed?
  4. In summary, SDRE has already been applied to this type of problem. So to make this paper more interesting and to extend the novelty the authors should explain how their approach is different to the aforementioned paper. In addition, the actuator configurations should be described, and the control allocation strategy explained too. The paper lacks some detail. More effort could be given to expanding the literature review and explaining the results.

Round 2

Reviewer 3 Report

The author's have done a better job with the literature review and I think it is acceptable. There is little novelty in terms of control design with respect to previous work, but there is some interesting analysis.